# Usefulness of muscle ultrasound in appendicular skeletal muscle mass estimation for sarcopenia assessment

**Seol-Hee Baek**[1], **Joo Hye Sung**[1], **Jin-Woo Park**[1], **Myeong Hun Son**[1], **Jung Hun Lee**[1], **Byung-Jo Kim**[1,2]*

**1** Department of Neurology, Korea University Anam Hospital, Seoul, Republic of Korea, **2** BK21 FOUR Program in Learning Health Systems, Korea University, Seoul, South Korea

* nukbj@korea.ac.kr

**Data Availability Statement:** All relevant data are within the paper and its Supporting Information files.

## Abstract

The measurement of skeletal muscle mass is essential for the diagnosis of sarcopenia. Muscle ultrasonography has emerged as a useful tool for evaluating sarcopenia because it can be used to assess muscle quality and quantity. This study investigated whether muscle ultrasonography is effective for estimating appendicular skeletal muscle mass (ASM) and screening for sarcopenia. This study prospectively enrolled 212 healthy volunteers aged 40–80 years. ASM was measured using the bioelectrical impedance analysis. Muscle thickness (MT) and echo-intensity (EI) were measured in four muscles (biceps brachii, BB; triceps brachii, TB; rectus femoris, RF; biceps femoris, BF) on the dominant hand. A hold-out cross-validation method was used to develop and validate the ASM prediction equation. In the model development group, the ASM prediction equations were deduced as follows: estimated ASM for men (kg) = 0.167 × weight (kg) + 0.228 × height (cm) + 0.143 × MT of BF (mm)– 0.822 × EI to MT ratio of BB– 28.187 ($R^2$ = 0.830) and estimated ASM for women (kg) = 0.115 × weight + 0.215 × height (cm) + 0.139 × MT of RF–0.638 × EI to MT ratio of BB– 23.502 ($R^2$ = 0.859). In the cross-validation group, the estimated ASM did not significantly differ from the measured ASM in both men (p = 0.775; intraclass correlation coefficient [ICC] = 0.948) and women (p = 0.516; ICC = 0.973). In addition, multiple logistic regression analysis revealed that the ratios of EI to MT in the BF and RF muscles in men and MT in the BB muscle in women could be valuable parameters for sarcopenia screening. Therefore, our study suggests that muscle ultrasound could be an effective tool for estimating ASM and screening sarcopenia.

## Introduction

Sarcopenia is a skeletal muscle disorder characterized by progressive and generalized loss of muscle mass and strength and reduced functional physical performance [1]. This could be related to the increased risk of falls, frailty, and mortality. Recently, sarcopenia has been recognized as an independent disease condition with a dedicated International Classification of Disease-10 code and not just an age-related decline in muscle mass [2]. The European Working

**Funding:** This work was supported by the Industrial Technology Innovation Program (No. 20008842), funded by the Ministry of Trade, Industry & Energy (MOTIE, Korea). The funders had no role in study design, data collection and analysis, decision to publish, or preparation of the manuscript.

**Competing interests:** The authors have declared that no competing interests exist.

Group on Sarcopenia in Older People (EWGSOP) and the Asian Working Group for Sarcopenia (AWGS) recommended that the diagnosis of sarcopenia should be based on the combination of muscle mass, muscle strength, and physical performance [3, 4]. Therefore, measurement of muscle mass is an important part of sarcopenia diagnosis. The EWGSOP and AWGS suggested sarcopenia cut-off points for low muscle mass using appendicular skeletal muscle mass (ASM), which is a sum of the muscle mass in the limbs, or height-adjusted ASM (ASM/height$^2$) [3, 4]. Muscle mass has been measured by several techniques, including dual-energy X-ray absorptiometry (DEXA), bioelectrical impedance analysis (BIA), computer tomography (CT), and magnetic resonance imaging (MRI). MRI and CT are considered the gold standards for the assessment of muscle mass because of their high accuracy. However, these techniques are not commonly used in clinical practice because of their high cost. In contrast, the DEXA and BIA techniques are widely used for the assessment of muscle mass because they are relatively inexpensive and easy to use.

Recently, ultrasound has been widely used as an effective neuroimaging tool for assessing the peripheral nerves and muscles. Muscle ultrasound can evaluate muscle thickness, cross-sectional area, muscle architecture, and muscle echo-intensity (EI). Therefore, muscle ultrasound may have the potential to evaluate both muscle quantity and quality. Although several studies have reported the possibility of muscle mass measurement using muscle ultrasound [5–9], it is still not approved as a neuroimaging tool for assessing muscle mass according to the EWGOP and AWGS recommendations [3, 4]. In addition, whether muscle EI is valuable for assessing muscle mass and further investigating sarcopenia is rarely reported.

This study aimed to investigate whether muscle ultrasound can be used as an effective tool for estimating ASM in terms of muscle quantity and quality. Since muscle mass begins to decline in middle-aged, and decreases gradually with age [10], we explored whether muscle ultrasonography could be used as a screening tool for sarcopenia in middle-aged and older individuals.

## Materials and methods

### Study population

Healthy volunteers aged 41–80 years were prospectively recruited between October 2020 and December 2021. We excluded participants with any objectively detected muscle weakness at the time of the study, any neurological disorder or musculoskeletal disease that could have caused muscle weakness within recent 3 months, or gait disturbance or muscle weakness as a sequela of a previous neurological disorder or musculoskeletal disease. A hold-out cross-validation method was used to develop and validate the ASM prediction equation. Thus, the entire dataset of our study was randomly divided into a model development group (training set) and a validation group (testing set). The ASM prediction equation using ultrasound parameters was then deduced from the model development group, and the accuracy of deducing the ASM equation was verified in the cross-validation group. The enrolled participants were randomly divided into a model development group (70%) and a cross-validation group (30%) using SPSS 25.0 (IBM Corp., Armonk, NY, USA). Written informed consent was obtained from all participants prior to inclusion. This study was approved by the Institutional Review Board of Korea at the University Anam Hospital (No. 2020AN0361) and performed in accordance with the Declaration of Helsinki.

### Clinical assessment

We collected demographic and clinical information on the participants, including their age, sex, height, weight, and body mass index (BMI). To evaluate muscle strength, the handgrip

strength of the dominant hand was measured using hand-held dynamometry (Jamar hand dynamometry, TEC Inc., Clifton, NJ, USA). To evaluate physical performance, gait speed was measured using gait analysis equipment (GAITRite®, CIR Systems Inc., NJ, USA). To measure muscle mass, body composition analysis was performed via BIA methods using InBody770 (InBody Co. LTD, Seoul, Korea). The InBody 770 machine had a total of 30 impedance measurements at six different frequencies (1 kHz, 5 kHz, 50 kHz, 250 kHz, 500 kHz, and 1000 kHz) for five body segments (right and left arms, right and left legs, and trunk). Body composition was estimated according to the developed prediction equations using tissue conductivity variables in combination with other covariates such as sex, weight, and height. In addition, the participants were instructed not to eat or exercise for at least three hours before the test and to maintain normal fluid intake the day before the test. Body composition data, including ASM data, were collected. Since muscle mass correlates with body size, muscle mass-adjusted body size is required to identify the optimal cut-off point for sarcopenia. The EWGSOP and AWGS 2019 consensus have proposed the cutoff point of sarcopenia using ASM normalized with the squared height [3, 4]. Thus, ASM index was calculated using the following equation: ASM index $(kg/m^2)$ = ASM (kg)/ height $(m)^2$.

## Muscle ultrasound

Muscle ultrasound was performed using a diagnostic ultrasound device (Aplio i700, Canon Medical Systems, Tochigi, Japan) with an 18-MHz linear probe and B-mode scanning, which was set with Gain 78, a penetrance depth of 3.5 cm, and a probe length of 5.5 cm. In this study, two representative muscles from the upper and lower extremities that are easy to assess using ultrasound were selected. Thus, the biceps brachii (BB) and triceps brachii (TB) in the upper extremity and the rectus femoris (RF) and biceps femoris (BF) in the extremity were chosen for study. MT and EI of these muscles were measured on the dominant hand side. The probe was placed perpendicular to the skin at minimal pressure to ensure accurate measurements. All participants were asked to be fully relaxed while lying in a supine position for the examination of the BB, TB, and RF muscles and in a prone position for the BF muscle. MT was measured in the short-axis view at the maximal vertical distance from the superficial to deep fascia layers. For the BB and TB muscles, the probe was placed between the acromion and cubital crease and scanned along the BB muscle to identify the thickest point of the BB muscle. Subsequently, the probe was laterally rotated 90° to identify the thickest point of the TB muscle. In addition, the probe was placed at the midpoint between the anterior superior iliac spine and the superior border of the patella for the RF muscle, and at the midpoint between the ischial tuberosity and fibular head for the BF muscle, followed by scanning along the RF and BF muscles to identify the thickest point of each muscle. To measure EI, regions of interest were drawn at each muscle that included maximum muscle tissue without the bone or surrounding fascia. The EI in the region of interest of each muscle was measured thrice and averaged for analysis. Fig 1 shows the methods used to measure ultrasonography parameters in each muscle.

## Statistical analysis

Descriptive summaries are presented as frequencies and proportions for categorical variables and means and standard deviations for continuous variables. Pearson's correlation analysis was performed to demonstrate the correlation between clinical parameters and ASM. To develop an ASM prediction equation, a multiple linear regression analysis was performed in the model development group, and a paired t-test was performed to investigate the agreement between the measured and estimated ASM. In addition, the Bland–Altman plot was used to

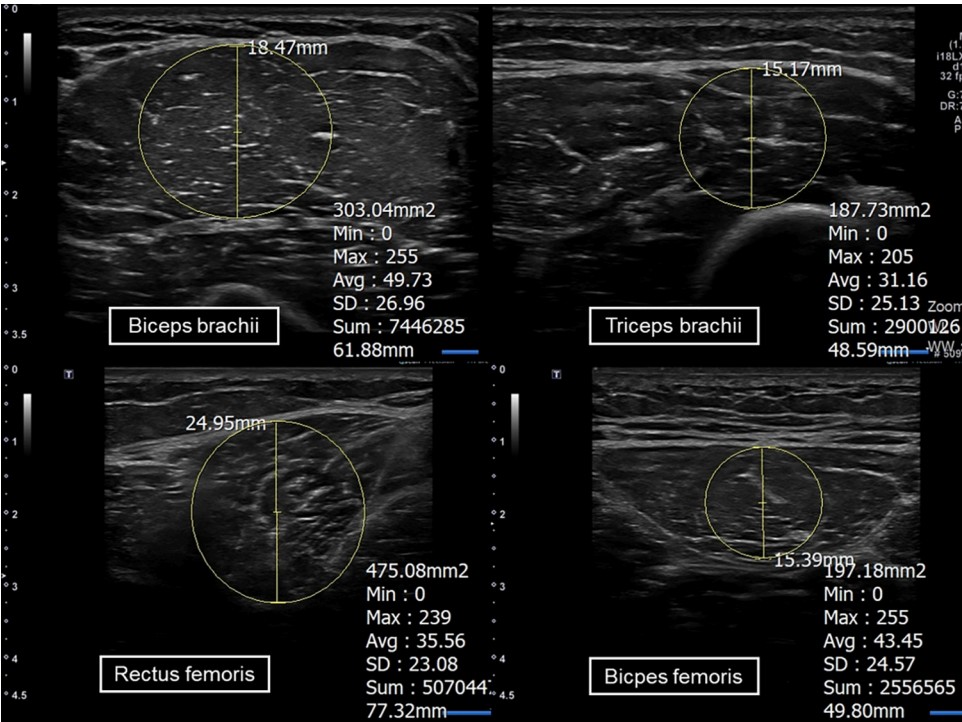

**Fig 1. Standard examination parameters.** Muscle thickness and echo intensity of the four selected muscles (biceps brachii, triceps brachii, rectus femoris, and biceps femoris muscles) on the dominant hand side using a muscle ultrasound. The muscle thickness was measured in the short-axis view at the maximal vertical distance from the superficial to deep fascia layers. To measure echo intensity, regions of interest were drawn at each muscle that included maximum muscle tissue without bone or surrounding fascia.

compare the BIA-measured and estimated ASM. The prediction equation of ASM derived from the model development group was applied to the cross-validation group. A paired t-test and two-way random-effect intraclass correlation coefficient (ICC) were used to test the agreement between the measured and estimated ASM in the cross-validation group. In addition, a multivariate logistic regression model was used to identify the potential risk factors for sarcopenia. Low ASM group was defined as, based on AWGS 2019 consensus, ASM index of $<7.0$ kg/m$^2$ for men and $<5.7$ kg/m$^2$ for women by employing BIA methods [4]. Statistical comparisons between the normal and low ASM groups were performed using independent t-tests for continuous variables. The discriminative power of each parameter was assessed using the receiver operating characteristic curve and area under the curve (AUC). Statistical significance was set at $p < 0.05$. All statistical analyses were performed using SPSS 25.0 (IBM Corp., Armonk, NY, USA).

## Results

### Study participants

This study included 214 healthy volunteers (91 men, 123 women), of which two were excluded: one with inadequate data and another with a history of poliomyelitis. Thus, 212 participants (91 men and 121 women) were finally enrolled in this study. Men (n = 63 and 28) and women (n = 82 and 39) were included in the model development and cross-validation groups, respectively. Descriptive clinical data and ultrasound parameters for each group are summarized in Table 1. Pearson's correlation analysis between the clinical parameters and ASM was

**Table 1. Demographic and clinical characteristics of the study participants.**

| | Total (n = 212) | Model development group | | Cross-validation group | |
|---|---|---|---|---|---|
| | | Men (n = 61) | Women (n = 90) | Men (n = 30) | Women (n = 31) |
| | Mean ± SD | Mean ± SD | Mean ± SD | Mean ± SD | Mean ± SD |
| **Age, years** | 59.47 ± 10.66 | 61.21 ± 11.40 | 57.88 ± 9.88 | 58.77 ± 12.40 | 61.35 ± 9.13 |
| **Height, cm** | 162.42 ± 8.38 | 169.53 ± 5.66 | 157.27 ± 5.79 | 169.58 ± 6.02 | 156.48 ± 4.98 |
| **Weight, kg** | 64.64 ± 11.22 | 72.10 ± 10.40 | 57.62 ± 7.19 | 73.63 ± 8.44 | 61.63 ± 9.77 |
| **BMI, kg/m$^2$** | 24.39 ± 3.03 | 25.04 ± 3.07 | 23.30 ± 2.63 | 25.56 ± 2.20 | 25.17 ± 3.74 |
| **Hand grip strength, kg** | 26.03 ± 8.68 | 33.36 ± 7.15 | 21.14 ± 4.84 | 32.67 ± 6.46 | 19.35 ± 6.85 |
| **Walking speed, cm/s** | 119.85 ± 18.47 | 119.12 ± 19.59 | 119.38 ± 18.28 | 121.52 ± 19.51 | 121.01 ± 16.30 |
| **ASM, kg** | 18.96 ± 4.55 | 23.10 ± 3.26 | 15.65 ± 2.05 | 23.61 ± 2.88 | 15.91 ± 2.26 |
| **ASM/Height$^2$, kg/m$^2$** | 7.09 ± 1.09 | 8.01 ± 0.84 | 6.31 ± 0.55 | 8.18 ± 0.63 | 6.48 ± 0.70 |
| **Muscle thickness** | | | | | |
| Biceps brachii, mm | 13.89 ± 3.04 | 16.36 ± 2.41 | 11.85 ± 1.75 | 16.42 ± 2.96 | 12.53 ± 1.59 |
| Triceps brachii, mm | 10.59 ± 3.51 | 12.48 ± 3.96 | 9.12 ± 2.39 | 12.42 ± 3.68 | 9.37 ± 2.58 |
| Rectus femoris, mm | 11.26 ± 2.51 | 12.73 ± 2.49 | 10.23 ± 1.95 | 12.53 ± 2.27 | 10.10 ± 2.23 |
| Biceps femoris, mm | 19.03 ± 4.53 | 21.34 ± 3.79 | 17.12 ± 3.88 | 22.03 ± 4.51 | 17.10 ± 4.09 |
| **Muscle echo intensity** | | | | | |
| Biceps brachii | 49.63 ± 2.50 | 48.84 ± 1.88 | 50.11 ± 2.96 | 48.39 ± 1.64 | 51.02 ± 1.62 |
| Triceps brachii | 34.99 ± 8.42 | 31.36 ± 8.10 | 37.75 ± 7.50 | 30.81 ± 8.29 | 38.17 ± 7.50 |
| Rectus femoris | 45.69 ± 3.96 | 45.37 ± 4.14 | 45.80 ± 3.63 | 45.52 ± 3.16 | 46.19 ± 5.16 |
| Biceps femoris | 40.51 ± 5.85 | 41.20 ± 5.49 | 40.81 ± 5.51 | 36.79 ± 7.49 | 41.84 ± 4.39 |

SD, standard deviation; BMI, body mass index; ASM, appendicular skeletal muscle mass.

performed. ASM was positively correlated with height, weight, BMI, and hand grip strength and negatively correlated with age. In addition, ASM was correlated with muscle thickness but negatively correlated with muscle EI. The results of the correlation analysis are summarized in Table 2.

## Development and validation of equation model for estimating ASM using muscle ultrasound parameters

Multiple linear regression was performed in the model development group to develop an equation for estimating the ASM using muscle ultrasound parameters. In the men group, weight, height, MT of the BF muscle, and the ratio of EI to MT of the BB muscle were predictors of ASM (Table 3 and S1 Table). The multiple linear regression model produced the following equation for estimated ASM in men:

*Estimated ASM for men (kg) = 0.167 × weight (kg) + 0.228 × height (cm) + 0143 × BF thickness (mm) − 0.822 × ratio of EI to MT of BB − 28.187* ($R^2$ = 0.830, adjusted $R^2$ = 0.818).

In the women group, weight, height, MT of the RF muscle, and the ratio of EI to MT in the BB muscle were predictors of ASM (Table 4 and S2 Table). The multiple linear regression model produced the following equation for estimated ASM in women:

*Estimated ASM for women (kg) = 0.115 × weight + 0.215 × height (cm) + 0.139 × RF thickness − 0.638 × ratio of EI to MT of BB – 23.502* ($R^2$ = 0.859; adjusted $R^2$ = 0.853; Fig 2B).

In the model development group, the estimated ASM did not significantly differ from the measured ASM in either group: men, p = 0.749; women, p = 0.548 (Fig 2), without a significant systematic error in the Bland–Altman plot (Fig 3). The developed ASM prediction equation was applied in the cross-validation group. The mean value of estimated ASM did not differ

**Table 2. Correlation analysis between clinical parameters and appendicular skeletal muscle mass.**

|  | Total (n = 212) | | Men (n = 91) | | Women (n = 121) | |
|---|---|---|---|---|---|---|
|  | r | p-value | r | p-value | r | p-value |
| Age, years | -0.184 | 0.007 | -0.527 | <0.001 | -0.319 | <0.001 |
| Height, cm | 0.889 | <0.001 | 0.740 | <0.001 | 0.751 | <0.001 |
| Weight, kg | 0.859 | <0.001 | 0.827 | <0.001 | 0.741 | <0.001 |
| BMI, kg/m² | 0.441 | <0.001 | 0.547 | <0.001 | 0.374 | <0.001 |
| Hand grip strength, kg | 0.758 | <0.001 | 0.470 | <0.001 | 0.380 | <0.001 |
| Walking speed, cm/s | -0.012 | 0.860 | -0.062 | 0.562 | 0.018 | 0.848 |
| Muscle thickness |  |  |  |  |  |  |
| Biceps brachii, mm | 0.718 | <0.001 | 0.327 | 0.002 | 0.339 | <0.001 |
| Triceps brachii, mm | 0.525 | <0.001 | 0.341 | 0.001 | 0.194 | 0.033 |
| Rectus femoris, mm | 0.553 | <0.001 | 0.361 | <0.001 | 0.234 | 0.010 |
| Biceps femoris, mm | 0.497 | <0.001 | 0.393 | <0.001 | -0.027 | 0.765 |
| Muscle echo intensity |  |  |  |  |  |  |
| Biceps brachii | -0.299 | <0.001 | -0.120 | 0.259 | -0.010 | 0.912 |
| Triceps brachii | -0.269 | <0.001 | 0.145 | 0.169 | 0.057 | 0.535 |
| Rectus femoris | -0.067 | 0.332 | -0.075 | 0.478 | 0.013 | 0.886 |
| Biceps femoris | -0.235 | 0.001 | -0.395 | <0.001 | -0.055 | 0.549 |

BMI, body mass index.

significantly from that of measured ASM in either men (23.61 kg vs 23.53 kg, p = 0.755; ICC = 0.948) or women (15.91 kg vs 15.99 kg, p = 0.516; ICC = 0.973) in the cross-validation group (Table 5).

## Cut-off value in ultrasound-derived parameters for sarcopenia risk screening

A low ASM index is essential for diagnosing sarcopenia. Among the study population, eight men and 13 women met the low ASM index criteria and were classified into the low ASM group. The low ASM group was older than the normal ASM group in men; however, no difference in age was observed between the two groups in women. Both men and women in the low ASM group exhibited weaker handgrip strength. Additionally, the low ASM group had a lower MT of the BB, TB, and RF muscles in men and the BB and TB muscles in women. However, there were no significant differences in the EI of any muscle between the two groups, except for the BF muscle in men. The clinical characteristics of the normal and low ASM groups are summarized in Table 6.

**Table 3. Regression models for predicting appendicular skeletal muscle mass in the men group.**

|  | r | $R^2$ | Adjusted $R^2$ | SEE | F | Sig. |
|---|---|---|---|---|---|---|
| Model 1 | 0.880 | 0.775 | 0.767 | 1.573 | 99.657 | <0.001 |
| Model 2 | 0.905 | 0.818 | 0.809 | 1.426 | 85.497 | <0.001 |
| Model 3 | 0.909 | 0.827 | 0.814 | 1.404 | 66.780 | <0.001 |
| Model 4 | **0.911** | **0.830** | **0.818** | **1.390** | **66.446** | **<0.001** |

SEE, standard error of the estimate.

Model 1: weight and height; Model 2: weight, height, and thickness of the biceps femoris; Model 3: weight, height, and echo intensity to muscle thickness ratio of the biceps brachii and rectus femoris; and Model 4: weight, height, thickness of the biceps femoris, and echo intensity to muscle thickness ratio of the biceps brachii.

**Table 4. Regression models for predicting appendicular skeletal muscle mass in the women group.**

| | r | $R^2$ | Adjusted $R^2$ | SEE | F | Sig. |
|---|---|---|---|---|---|---|
| Model 1 | 0.892 | 0.796 | 0.791 | 0.937 | 169.329 | <0.001 |
| Model 2 | 0.925 | 0.855 | 0.848 | 0.799 | 125.024 | <0.001 |
| Model 3 | 0.923 | 0.852 | 0.845 | 0.806 | 122.505 | <0.001 |
| Model 4 | **0.927** | **0.859** | **0.853** | **0.786** | **129.847** | **<0.001** |

SEE, standard error of the estimate.

Model 1: weight and height; Model 2: weight, height, and thickness of the biceps brachii and rectus femoris; Model 3: weight, height, and echo intensity to muscle thickness ratio of the biceps brachii and rectus femoris; Model 4: weight, height, thickness of the rectus femoris, and echo intensity to muscle thickness ratio of the biceps brachii.

Multivariate logistic regression analysis was performed to identify potential risk factors for sarcopenia (Table 7). In the men group, the EI to MT ratio of the RF muscle (odds ratio [OR] 3.536, 95% confidence interval [CI] 1.462–8.555; p = 0.005) and BF muscle (OR 3.2214, 95% CI 1.059–9.750; p = 0.039) were associated with low ASM. The EI to MT ratio of the RF and BF muscles for predicting the risk of low ASM had AUC of 0.770 (p = 0.012) and 0.803 (p = 0.005), respectively. The optimal cut-off points for predicting the risk of the low ASM group were an EI to MT ratio of ≥4.19 for RF and ≥2.14 for BF and BF. Using these cut-off values, a sensitivity of 75.0% and 87.5%, specificity of 75.9% and 73.5%, positive predictive value of 23.1% and 24.1%, and negative predictive value of 96.9% and 98.4% were obtained, respectively. In addition, the AUC of the combination of these two variables for predicting the risk of low ASM was 0.887 (p < 0.001). In the women group, the MT of the BB muscle (OR 0.644, 95% CI, 0.442–0.939; p = 0.022) was associated with low ASM. The MT of BB muscle for predicting the risk of low ASM had an AUC of 0.727 (p = 0.008), and the optimal cut-off value was BB ≤ 11.56 mm. Using this cut-off value, a sensitivity of 92.3%, specificity of 61.1%, positive predictive value of 22.2%, and negative predictive value of 98.4% were obtained.

## Discussion

This study deduced ultrasound-driven estimation equations of the ASM using a multiple linear regression model. Additionally, the estimated ASM was not significantly different from the

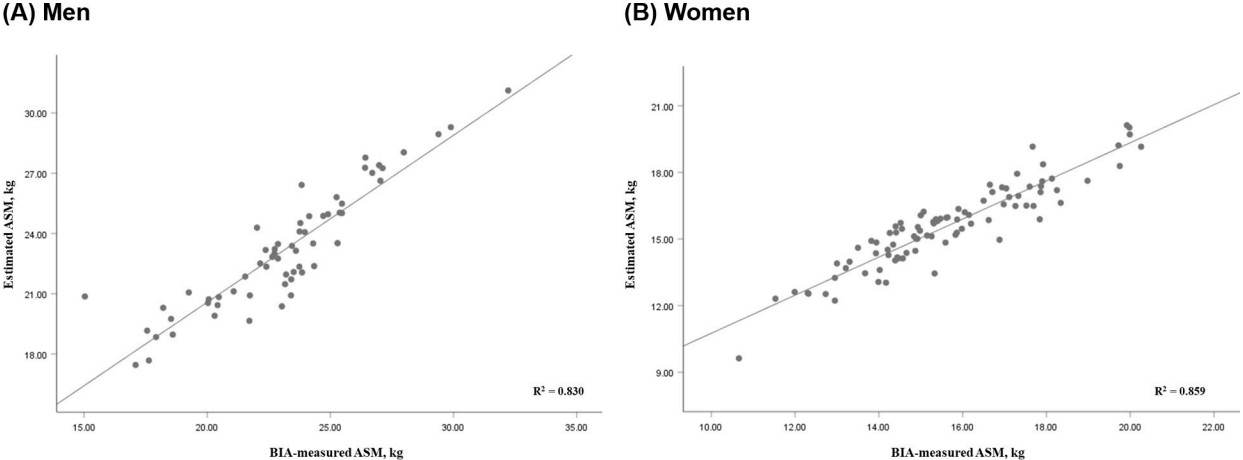

**(A) Men**

**(B) Women**

**Fig 2. Relationship between the BIA-measured ASM and estimated ASM in the model development group.** BIA, bioelectrical impedance analysis; ASM, appendicular skeletal muscle mass.

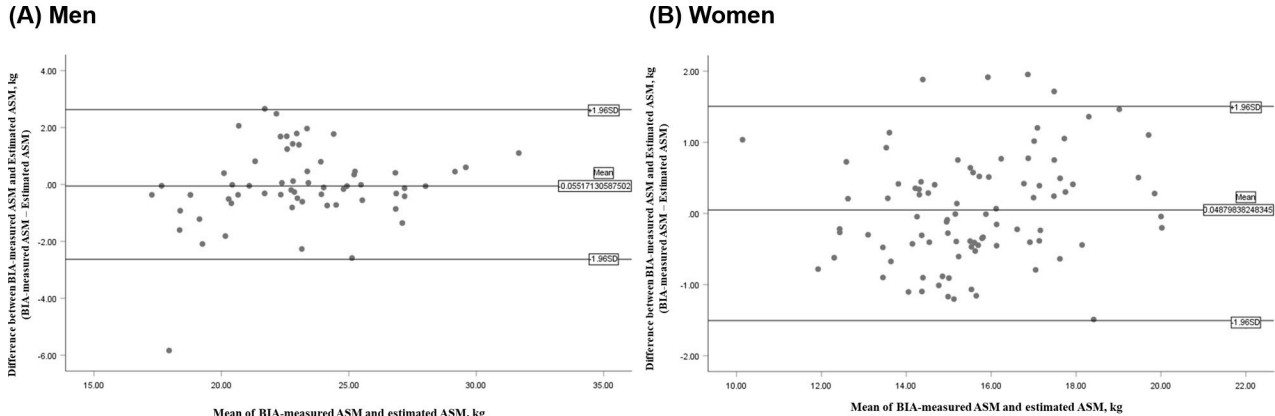

**Fig 3. Bland–Altman plot for the agreement between the BIA-measured ASM and estimated ASM in the model development group.** BIA, bioelectrical impedance analysis; ASM, appendicular skeletal muscle mass.

BIA-measured ASM in the men and women groups. Furthermore, our study showed that muscle ultrasound parameters, particularly the EI to MT ratio of the thigh muscles (BF and RF muscles) in the men group and the MT of the BB muscle in the women group, were associated with risk factors for low ASM.

Some studies have suggested that muscle ultrasonography may be a reliable tool for estimating the muscle mass. Abe et al. [11] developed a prediction equation for estimating lean body mass using MT that was measured by employing B-mode ultrasound. However, this study used the sum of the MT at nine sites (forearm, biceps, triceps, abdomen, subscapular, quadriceps, hamstrings, gastrocnemius, and tibialis anterior muscles) as the equation parameters. Thus, it is difficult and time-consuming to estimate the muscle mass. Takai et al. developed an estimation equation for whole-body fat-free mass using MT at four sites (anterior upper arm, anterior and posterior thigh, and posterior lower leg) measured using ultrasound and limb length [12]. Furthermore, some studies have reported that ASM can be predicted using a single ultrasound image of the forearm [6, 13]. Our study revealed that ASM can be effectively estimated by evaluating only two muscles in both men and women. This result suggests that muscle mass can be quickly and easily assessed using ultrasonography. In summary, muscle ultrasonography is a reliable and effective tool for estimating the muscle mass.

Some studies have reported that muscle ultrasonography could be used as a screening tool for sarcopenia. Rustani et al. suggested that RF thickness is a valuable parameter for sarcopenia screening [8]. Yamada et al. reported that quadriceps MT and thigh muscle volume can be indicators of sarcopenia diagnosis [14]. In addition, some studies suggested that the thickness of calf muscles, such as the tibialis anterior and gastrocnemius, can be a favorable parameter for sarcopenia screening [9, 15–17]. Some studies have also suggested that evaluation of arm muscles can help in screening for sarcopenia [6, 18, 19]. In this study, we found that muscle

**Table 5. Intraclass correlation coefficient and standard error of estimated appendicular skeletal muscle mass in the cross-validation group.**

|  | ICC (95% CI) | p-value | Standard error (95% CI) |
|---|---|---|---|
| Men | 0.948 (0.890–0.975) | <0.001 | 0.077 (-0.421–0.574) |
| Women | 0.973 (0.944–0.987) | <0.001 | 0.124 (-0336–0.172) |

ASM, appendicular skeletal muscle mass; CI, confidence interval; ICC, intraclass correlation coefficient.

**Table 6. Comparison between subjects in the normal and low ASM groups.**

| | Men | | | Women | | |
|---|---|---|---|---|---|---|
| | ASM ≥ 7.0 | ASM < 7.0 | p-value | ASM ≥ 5.7 | ASM < 5.7 | p-value |
| **Number of patients** | 83 | 8 | | 108 | 13 | |
| **Age, years** | 59.27 ± 11.53 | 72.25 ± 5.82 | **0.002** | 58.77 ± 9.88 | 58.77 ± 9.24 | 0.933 |
| **Height, cm** | 170.03 ± 5.68 | 164.58 ± 3.85 | **0.010** | 157.67 ± 5.12 | 152.06 ± 6.94 | **0.008** |
| **Weight, kg** | 74.00 ± 8.92 | 58.06 ± 5.68 | **0.000** | 59.76 ± 7.76 | 49.37 ± 3.27 | **0.000** |
| BMI, kg/m$^2$ | 25.58 ± 2.63 | 21.41 ± 1.66 | **0.000** | 24.06 ± 3.04 | 21.44 ± 2.09 | **0.001** |
| **Hand grip strength, kg** | 33.87 ± 6.42 | 25.50 ± 7.54 | **0.003** | 20.88 ± 5.36 | 19.00 ± 6.12 | 0.395 |
| **Walking speed, cm/s** | 118.71 ± 19.05 | 132.44 ± 20.89 | 0.128 | 119.45 ± 17.46 | 122.75 ± 20.47 | 0.815 |
| **ASM, kg** | 23.83 ± 2.64 | 17.44 ± 1.36 | 0.000 | 16.08 ± 1.87 | 12.68 ± 1.27 | 0.000 |
| ASM/Height$^2$, kg/m$^2$ | 8.22 ± 0.60 | 6.44 ± 0.53 | 0.000 | 6.46 ± 0.53 | 5.47 ± 0.16 | 0.000 |
| **Muscle thickness** | | | | | | |
| Biceps brachii, mm | 16.57 ± 2.61 | 14.39 ± 1.21 | **0.005** | 12.15 ± 1.77 | 10.94 ± 0.75 | **0.008** |
| Triceps brachii, mm | 12.76 ± 3.85 | 9.32 ± 2.18 | **0.011** | 9.34 ± 2.51 | 7.88 ± 1.05 | **0.024** |
| Rectus femoris, mm | 12.91 ± 2.29 | 10.11 ± 2.27 | **0.004** | 10.30 ± 2.05 | 9.32 ± 1.56 | 0.085 |
| Biceps femoris, mm | 21.80 ± 4.04 | 19.10 ± 3.09 | 0.051 | 17.26 ± 3.90 | 15.93 ± 4.03 | 0.252 |
| **Muscle echo intensity** | | | | | | |
| Biceps brachii | 48.64 ± 1.81 | 49.15 ± 1.89 | 0.585 | 50.39 ±2.79 | 49.99 ± 1.88 | 0.359 |
| Triceps brachii | 31.18 ± 8.41 | 31.14 ± 4.37 | 0.911 | 38.02 ± 7.19 | 36.51 ± 9.77 | 0.973 |
| Rectus femoris | 45.32 ± 3.76 | 46.42 ± 4.63 | 0.231 | 45.95 ± 4.01 | 45.43 ± 4.55 | 0.586 |
| Biceps femoris | 39.16 ± 6.39 | 45.82 ± 4.65 | **0.005** | 41.02 ± 5.28 | 41.52 ± 5.16 | 0.642 |
| **EI to MT ratio** | | | | | | |
| Biceps brachii | 3.01 ± 0.54 | 3.43 ± 0.25 | **0.005** | 4.23 ± 0.65 | 4.59 ± 0.40 | **0.025** |
| Triceps brachii | 2.75 ± 1.23 | 3.49 ± 0.91 | 0.077 | 4.38 ± 1.42 | 4.75 ± 1.45 | 0.148 |
| Rectus femoris | 3.64 ± 0.80 | 4.81 ± 1.31 | **0.012** | 4.68 ± 1.23 | 5.05 ± 1.26 | 0.344 |
| Biceps femoris | 1.89 ± 0.62 | 2.45 ± 0.42 | **0.005** | 2.54 ± 0.85 | 2.81 ± 0.92 | 0.255 |

Descriptive summaries are presented as means ± standard deviations for continuous variables.

ASM, appendicular skeletal muscle mass; BMI, body mass index; EI, echo intensity; MT, muscle thickness.

ultrasonography parameters, the EI to MT ratio of thigh muscles (RF and BF muscles) in men and MT of BB muscle in women, are associated with low ASM. These findings suggest that muscle ultrasonography is an easy and rapid screening method for sarcopenia. In addition, the results of this study imply that different muscles can be used as helpful parameters for sarcopenia screening according to sex. Further studies with a larger number of participants are needed to determine the appropriate muscle selection and optimal cut-off value.

Muscles (mainly contractile proteins) appear hypoechogenic, whereas adipocytes and fibrous tissue components appear hyperechogenic. Muscle EI is the degree of brightness of the acquired image and is affected by intramuscular fat infiltration and fibrotic changes, which are key factors in determining muscle function quality [20]. Therefore, muscle EI reflects the proportion of adipocytes and fibrous tissue in muscles [21, 22] and is considered a parameter that could reflect muscle quality. Some studies reported that muscle EI may be associated with muscle strength and functional status [23–25]. Previous research has reported that muscle EI is negatively correlated with MT in older adults [26]. Another study reported that changes in muscle EI were significantly associated with changes in BMI [27]. These findings imply that muscle EI has a more complicated relationship with MT and other demographic factors. However, few studies have reported an association between EI and muscle mass. This study showed no significant differences in the EI value of each muscle between the normal and low ASM

**Table 7. Univariate and multivariate logistic regression analyses.**

| Men group | | |
| --- | --- | --- |
| **Univariate logistic regression** | | |
| | **OR (95% CI)** | **p-value** |
| Muscle thickness | | |
| Biceps brachii | 0.690 (0.498–0.956) | 0.026 |
| Triceps brachii | 0.700 (0.511–0.957) | 0.026 |
| Rectus femoris | 0.527 (0.342–0.812) | 0.004 |
| Biceps femoris | 0.846 (0.703–1.018) | 0.076 |
| Muscle echo intensity | | |
| Biceps brachii | 1.169 (0.779–1.754) | 0.452 |
| Triceps brachii | 0.999 (0.9136–1.093) | 0.987 |
| Rectus femoris | 1.097 (0.870–1.383) | 0.433 |
| Biceps femoris | 1.20 (1.058–1.526) | 0.011 |
| EI to MT ratio | | |
| Biceps brachii | 3.562 (1.034–12.274) | 0.044 |
| Triceps brachii | 1.662 (0.891–3.099) | 0.110 |
| Rectus femoris | 3.409 (1.496–7.766) | 0.004 |
| Biceps femoris | 2.955 (1.051–8.305) | 0.040 |
| **Multivariate logistic regression (Forward stepwise)** | | |
| | **OR (95% CI)** | **p-value** |
| EI to MT ratio | | |
| Rectus femoris | 3.536 (1.462–8.555) | 0.005 |
| Biceps femoris | 3.214 (1.059–9.750) | 0.039 |
| **Women group** | | |
| **Univariate logistic regression** | | |
| | **OR (95% CI)** | **p-value** |
| Muscle thickness | | |
| Biceps brachii | 0.644 (0.442–0.939) | 0.022 |
| Triceps brachii | 0.759 (0.578–0.996) | 0.046 |
| Rectus femoris | 0.781 (0.582–1.049) | 0.100 |
| Biceps femoris | 0.916 (0.789–1.063) | 0.248 |
| Muscle echo intensity | | |
| Biceps brachii | 0.952 (0.787–1.152) | 0.614 |
| Triceps brachii | 0.974 (0.902–1.050) | 0.490 |
| Rectus femoris | 0.969 (0.842–1.115) | 0.657 |
| Biceps femoris | 1.019 (0.911–1.140) | 0.743 |
| EI to MT ratio | | |
| Biceps brachii | 2.429 (0.963–6.129) | 0.060 |
| Triceps brachii | 1.194 (0.803–1.774) | 0.382 |
| Rectus femoris | 1.245 (0.817–1.897) | 0.308 |
| Biceps femoris | 1.375 (0.755–2.504) | 0.298 |
| **Multivariate logistic regression (Forward stepwise)** | | |
| | **OR (95% CI)** | **p-value** |
| Muscle thickness | | |
| Biceps brachii | 0.644 (0.442–0.939) | 0.022 |

OR, odds ratio; CI, confidence interval; EI, echo intensity; MT, muscle thickness.

groups, except for the BF muscle in men. However, the present study showed that the EI to MT ratio was significantly associated with the estimation of muscle mass. These findings imply that the EI to MT ratio may be a more helpful parameter for sarcopenia assessment than the EI alone. Further studies are required to clarify the clinical implications of EI in the muscles.

This study had some limitations. First, this study had a very small number of participants with low ASM to obtain a cut-off value for sarcopenia screening. This study was designed to assess the reliability and usefulness of muscle ultrasonography in measuring ASM in healthy volunteers. Outcomes of healthy volunteers who met the sarcopenia criteria were used to validate the results of this study. Therefore, further studies with a larger number of participants with low ASM are needed to clarify the optimal cut-off value for sarcopenia screening. Second, ultrasound is an operator-dependent technique that is affected by different device settings. Additionally, the EI value of the muscle may differ according to the ultrasound device used. Therefore, to overcome this limitation, this study was designed for using a single ultrasound device. Accordingly, the equation developed for estimating ASM from the present study has been limited to various ultrasound devices. Nevertheless, the results of this study support the idea that muscle ultrasound could be an effective tool for estimating muscle mass. Further studies using various ultrasound devices are needed to clarify the optimal cut-off value of muscle EI for sarcopenia screening according to different ultrasound device. Third, muscle ultrasound data from the distal parts of the arm and leg were not included in this study. Since this study was designed to investigate a simple and quick equation for estimating ASM using muscle ultrasound, only the representative muscles that are easy to access via muscle ultrasound were selected.

## Conclusion

Muscle ultrasonography seems to be an effective tool for estimating muscle mass and screening for sarcopenia. Among the various parameters, MT and EI to MT ratio may be helpful indicators for assessing sarcopenia.

## Supporting information

**S1 Table. Multiple linear regression analysis (model 4) in men group.**
(DOCX)

**S2 Table. Multiple linear regression analysis (model 4) in women group.**
(DOCX)

## Acknowledgments

We would like to thank our colleagues in the Department of Neurology, Korea University Anam Hospital for their enthusiastic assistance. We also thank Kyoung-Sook Yang for providing statistical advice and analysis.

## Author Contributions

**Conceptualization:** Seol-Hee Baek, Joo Hye Sung, Byung-Jo Kim.

**Data curation:** Seol-Hee Baek, Joo Hye Sung, Jin-Woo Park, Myeong Hun Son, Jung Hun Lee.

**Formal analysis:** Seol-Hee Baek.

**Funding acquisition:** Byung-Jo Kim.

**Investigation:** Seol-Hee Baek, Joo Hye Sung, Jin-Woo Park, Myeong Hun Son, Jung Hun Lee.

**Methodology:** Byung-Jo Kim.

**Project administration:** Byung-Jo Kim.

**Supervision:** Byung-Jo Kim.

**Validation:** Seol-Hee Baek.

**Visualization:** Seol-Hee Baek.

**Writing – original draft:** Seol-Hee Baek.

**Writing – review & editing:** Byung-Jo Kim.

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
