## [Decision Letter · Decision Letter 0]

12 Sep 2022

PONE-D-22-16915Usefulness of muscle ultrasound in appendicular skeletal muscle mass estimation for sarcopenia assessmentPLOS ONE

Dear Dr. Kim,

Thank you for submitting your manuscript to PLOS ONE. After careful consideration, we feel that it has merit but does not fully meet PLOS ONE’s publication criteria as it currently stands. Therefore, we invite you to submit a revised version of the manuscript that addresses the points raised during the review process.

ACADEMIC EDITOR:Dear Authors, two experts in the field reviewed your manuscript and found several criticisms you should address during the revision process.

We look forward to receiving your revised manuscript.

Kind regards,

Emiliano Cè

Academic Editor

PLOS ONE

Journal Requirements:

This work was supported by the Industrial Technology Innovation Program (No. 20008842), funded by the Ministry of Trade, Industry & Energy (MOTIE, Korea). The funders had no role in study design, data collection and analysis, decision to publish, or preparation of the manuscript.

However, funding information should not appear in the Acknowledgments section or other areas of your manuscript. We will only publish funding information present in the Funding Statement section of the online submission form. 

This work was supported by the Industrial Technology Innovation Program (No. 20008842), funded by the Ministry of Trade, Industry & Energy (MOTIE, Korea). The funders had no role in study design, data collection and analysis, decision to publish, or preparation of the manuscript.

Reviewers' comments:

Reviewer's Responses to Questions

**Comments to the Author**

1. Is the manuscript technically sound, and do the data support the conclusions?

Reviewer #1: Yes

Reviewer #2: No

2. Has the statistical analysis been performed appropriately and rigorously? 

Reviewer #1: Yes

Reviewer #2: N/A

3. Have the authors made all data underlying the findings in their manuscript fully available?

Reviewer #1: Yes

Reviewer #2: Yes

4. Is the manuscript presented in an intelligible fashion and written in standard English?

Reviewer #1: Yes

Reviewer #2: Yes

5. Review Comments to the Author

Reviewer #1: Thanks to the authors for their good job. Sarcopenia is an important issue in aging societies and increasing comorbidities. This article could be a fine guide for the detection and management of sarcopenia. In this respect, I think it will contribute to the literature.

Reviewer #2: Dear Editor,

the manuscript entitled "Usefulness of muscle ultrasound in appendicular skeletal muscle mass estimation for

sarcopenia assessment" explore the use of ultrasound (US) as a simple, cheap, and non-invasive tool to assess sarcopenia.

To this purpose, 212 participants have been enrolled to the study.

I found the reading fluent and the english correct, however, I have some major concerns about the paper content.

1. Aiming to investigate whether muscle ultrasonography is an effective tool for estimating the appendicular skeletal muscle mass (ASM), the authors combined some parameters, among which some US measurements, into two equations (one for men and one for women). Then, they compared the estimated ASM to the one determined by a BIA analysis.

I found this approach questionable as BIA is not the gold standard for ASM assessment. Indeed, ASM is not a BIA direct measure but has to be derived by equation, therefore, its accuracy depends on which equation is adopted by the device. Moreover, other factors influence BIA measurements like hydration status, age, and body mass. About this latter, it has been found that BIA tends to underestimate ASM in overweight people. Even though the equations developed by the authors to estimate ASM took into account age, sex, and body mass, the measure used as reference value cannot be considered equal to the one determined by DXA or CT or MRI. In addition, ASM was estimated by authors feeding the equations with EI values determined during US measurements. Such index, however, has to be considered with cautions as its reliability is age-dependent (Strasser 2013) and it is influenced by several factors among which hydration status, probe position, and probe settings. Even though the reliability of the measure was good, the equations developed by the authors and used to estimate ASM, can be considered valid only in this specific study, with their specific US device, its settings and their operator. These equations on different laboratories may perform differently.

2. The other limit I found in the study relates to the US ability to discriminate lowASM people. Indeed, after having determined ASM on 212 participants, the authors found that 21 showed low ASM values and, by a ROC analysis, determined a cut-off value on the MT of some muscles. In my opinion the sample size of low ASM group was too poor to return a reliable value.

MINOR

1. In my opinion the introduction is too vague and should be improved, taking care in the terminology. The rationale is not clear and the paragraph dealing with sarcopenia should be delved more deeply. A complete definition of sarcopenia should be given: "Sarcopenia is a progressive, generalized, age-related loss of skeletal muscle mass [1]." is reductive. Notwithstanding the authors continued citing the association to loss of muscle strength and the reduction of physical activity, the offical definition of sarcopenia considers sarcopenia as concurrent reduction of skeletal muscle mass, muscle strength and functional physical performance. This latter is different from "physical activity".

"Historically, muscle mass has been measured using several imaging techniques, including dualenergy

X-ray absorptiometry, computed tomography, and magnetic resonance imaging." Which of these technique is considered the gold standard?

"Muscle ultrasound is a noninvasive, cost-effective, and easily accessible imaging technique for the evaluation of neuromuscular disorders [5]." I find this sentence unnecessary.

"This study aimed to investigate whether muscle ultrasonography is an effective tool for estimating the appendicular skeletal muscle mass (ASM) in terms of muscle quantity and quality." ASM should be introduced previously, in sarcopenia paragraph otherwise it is not clear why it represents a variable of interest. Moreover, given that the authors previously reported "muscle ultrasonography has been suggested as a valuable tool for assessing sarcopenia because it can assess both muscle quantity and quality" it is not clear which is the novelty of the present study. It seems this study aims to confirm something that is already known.

"In addition, we explored whether muscle ultrasonography can be used as a screening tool for sarcopenia in middle-aged and older individuals" please, explain why you chose such target of population. Is there any reason for involving middle-aged and old individuals?

"A hold-out cross-validation method was used to develop and validate the ASM prediction equation." Why did the authors have to validate an ASM prediction equation? The reader has to be previously be informed that US-derived ASM relies on prediction equation. Moreover, the authors should explain the reason why they proposed a different equation.

Please, do not use WEIGHT but BODY MASS

Please, provide an explanation about the test administered in Clinical assessment. Why did they test handgrip and gait speed, and analyzed body composition?

Pleae explain why ASM has been normalized to height.

Please, explain the criteria adopted to choose the muscles that have been measured (why BB, TB, RF ad BF)?

Please, explain why calf muscles have not been considered.

Please, it is recommended to provide all the information about US measurements: gain, frequency of the transducer beam, depth of penetrance, probe length.

As ROI appears only two times, I suggest to remove the acronymus.

In the statistical analysis, did the authors check for differences between DEVELOPMENT and cross-validation GROUP?

"Low ASM group was defined as, based on AWGS 2019 consensus, ASM index < 7.0 kg/m2 for men and < 5.7 kg/m2 for women." Please, provide a reference

Being sarcopenia a combination of muscle mass loss, low muscle strength and reduced functionality, did the authors investigate the correlation between functional parameters and ASM values?

Table 1 To make it more readable, I suggest to use only one column to show mean+/- SD and to thicken the column border of Model development goup and cross-validation group

Please, check that measure units are always present.

"The estimated ASM did not significantly differ from the measured ASM in either the men or the women groups (p = 0.749 and p = 0.548, respectively; Fig. 2)" Does this sentence relate to MODEL DEVELOPMENT group?

"without a significant systematic error in the Bland–Altman plot (p = 0.091 and p = 0.056, respectively; Fig. 3)." from which analysis, do these values derive?

Table 4. Differences in the measured and estimated appendicular skeletal muscle mass in the cross-validation group" I would appreciate a Bland_altman plot of these data.

About "Cut-off value in ultrasound-derived parameters for sarcopenia screening" I found questionable diagnosing sarcopenia on ASM alone. Similarly, I found questionable performing statistical analysis on samples with such diversity in size.

"Table 5. Comparison between subjects in the normal and low ASM groups." These results come from a statistical analysis that compared groups of different size (within male comparison: 83 vs 8 particpants; within female comparison: 108 vs 13 participants). There is no mention in statistical analysis paragraph about this comparison and, in any case, the size is so different that the results lose validity.

Please add references to:

"In addition, muscle ultrasonography has been suggested as a valuable tool for assessing sarcopenia because it can assess both muscle quantity and quality."

"Low ASM group was defined as, based on AWGS 2019 consensus, ASM index < 7.0 kg/m2 for men and < 5.7 kg/m2 for women"(reference is lacking)

6. PLOS authors have the option to publish the peer review history of their article (what does this mean?). If published, this will include your full peer review and any attached files.

Reviewer #1: No

Reviewer #2: No

---

## [Author Response · Author response to Decision Letter 0]

8 Nov 2022

On behalf of all the co-authors, I am very grateful to all the reviewers for their careful review, insightful comments, and constructive suggestions, which greatly helped to improve our manuscript. I have revised the manuscript to elaborate on and clarify the issues raised by the reviewers. I believe that the manuscript has substantially improved with the revision. 

The attached file named "Response to Reviewers" are our responses to the reviewers' comments, including how and where the text has been modified. I have highlighted the changes to the manuscript by using the track changes mode in MS Word. I hope that you find our responses satisfactory and that the manuscript is now acceptable for publication.

---

## [Decision Letter · Decision Letter 1]

6 Dec 2022

PONE-D-22-16915R1Usefulness of muscle ultrasound in appendicular skeletal muscle mass estimation for sarcopenia assessmentPLOS ONE

Dear Dr. Kim,

Thank you for submitting your manuscript to PLOS ONE. After careful consideration, we feel that it has merit but does not fully meet PLOS ONE’s publication criteria as it currently stands. Therefore, we invite you to submit a revised version of the manuscript that addresses the points raised during the review process.

ACADEMIC EDITOR: Two experts in the field reviewed your manuscript. They both found your new manuscript version decisively improved. Still Reviewer 2 has raised some minors issues that you should consider while revising the manuscript.==============================

We look forward to receiving your revised manuscript.

Kind regards,

Emiliano Cè

Academic Editor

PLOS ONE

Journal Requirements:

Reviewers' comments:

Reviewer's Responses to Questions

**Comments to the Author**

1. If the authors have adequately addressed your comments raised in a previous round of review and you feel that this manuscript is now acceptable for publication, you may indicate that here to bypass the “Comments to the Author” section, enter your conflict of interest statement in the “Confidential to Editor” section, and submit your "Accept" recommendation.

Reviewer #1: All comments have been addressed

Reviewer #2: (No Response)

2. Is the manuscript technically sound, and do the data support the conclusions?

Reviewer #1: Yes

Reviewer #2: Yes

3. Has the statistical analysis been performed appropriately and rigorously? 

Reviewer #1: Yes

Reviewer #2: Yes

4. Have the authors made all data underlying the findings in their manuscript fully available?

Reviewer #1: Yes

Reviewer #2: Yes

5. Is the manuscript presented in an intelligible fashion and written in standard English?

Reviewer #1: Yes

Reviewer #2: Yes

6. Review Comments to the Author

Reviewer #1: (No Response)

Reviewer #2: Dear Authors,

I appreciate your efforts in addressing most of my comments. I just have few minor concerns that I'd like to put to your attention.

In some comments I asked you to kindly explain your choices (comment 7, 8, 10, 11, and 12). You carefully addressed the comment in the Response section but, if I've seen correctly, you did not insert the explanation in the manuscript. My recommendations did not aim at receiving an explanation for myself rather providing more details to the readers by adding further explanantions in the text. Most of the times I had the answer so I was asking you to improve the manuscript by going deep insight for the readers.

Table 1 is almost ok. I personally do not appreciate decimals when they lose sense. For example: are two decimals essentials when the order of the measure unit is centimeters or millimeters? I understand that the table appears neater when every value is reported with two decimals but I wonder if the second decimals of a millimeter-based value has any sense.

In Conclusion I'd suggest a mitigation of the sentence "Muscle ultrasonography can be an effective tool for....." with something like "Muscle ultrasonography seems/appears be an effective tool for..."

7. PLOS authors have the option to publish the peer review history of their article (what does this mean?). If published, this will include your full peer review and any attached files.

Reviewer #1: No

Reviewer #2: No

---

## [Author Response · Author response to Decision Letter 1]

9 Dec 2022

Reviewers' comments

Reviewer #1: (No Response)

Reviewer #2: Dear Authors,

I appreciate your efforts in addressing most of my comments. I just have few minor concerns that I'd like to put to your attention.

Comments (1)

In some comments I asked you to kindly explain your choices (comment 7, 8, 10, 11, and 12). You carefully addressed the comment in the Response section but, if I've seen correctly, you did not insert the explanation in the manuscript. My recommendations did not aim at receiving an explanation for myself rather providing more details to the readers by adding further explanations in the text. Most of the times I had the answer so I was asking you to improve the manuscript by going deep insight for the readers.

Response (1)

We appreciate your careful reading of this manuscript and valuable comments on this research. We revised the manuscript, as you recommended.

1) About your previous comment #7, we modified the following sentence to clarify the purpose of our study and to explain why we included only healthy volunteers aged 41 – 80 years. 

(Line 70-73) Since muscle mass begins to decline in middle-aged, and decreases gradually with age [10], we explored whether muscle ultrasonography could be used as a screening tool for sarcopenia in middle-aged and older individuals.

2) About comment #8, we added the following sentences to describe a more detailed hold-out cross-validation method in the revised manuscript.

(Line 82-86) the entire dataset of our study was randomly divided into a model development group (training set) and a validation group (testing set). The ASM prediction equation using ultrasound parameters was then deduced from the model development group, and the accuracy of deducing the ASM equation was verified in the cross-validation group.

3) About comment #10-12, we modified the manuscript to clarify the methods of this study as follows: 

(Line 95-100) To evaluate muscle strength, the handgrip strength of the dominant hand was measured using hand-held dynamometry (Jamar hand dynamometry, TEC Inc., Clifton, NJ, USA). To evaluate physical performance, gait speed was measured using gait analysis equipment (GAITRite®, CIR Systems Inc., NJ, USA). To measure muscle mass, body composition analysis was performed via BIA methods using InBody770 (InBody Co. LTD, Seoul, Korea).

(Line 107-110) Since muscle mass correlates with body size, muscle mass-adjusted body size is required to identify the optimal cut-off point for sarcopenia. The EWGSOP and AWGS 2019 consensus have proposed the cutoff point of sarcopenia using ASM normalized with the squared height [3, 4]. 

(Line 116-123) In this study, two representative muscles from the upper and lower extremities that are easy to assess using ultrasound were selected. Thus, the biceps brachii (BB) and triceps brachii (TB) in the upper extremity and the rectus femoris (RF) and biceps femoris (BF) in the extremity were chosen for study. MT and EI of these muscles were measured on the dominant hand side.

Comments (2)

Table 1 is almost ok. I personally do not appreciate decimals when they lose sense. For example: are two decimals’ essentials when the order of the measure unit is centimeters or millimeters? I understand that the table appears neater when every value is reported with two decimals but I wonder if the second decimals of a millimeter-based value have any sense.

Response (2)

In Table 1, all variables were presented as two decimals. Although two decimals may not be essential when the order of the measuring unit is centimeters or millimeters, we presented the two decimals for a unified representation.

Comments (3)

In Conclusion I'd suggest a mitigation of the sentence "Muscle ultrasonography can be an effective tool for....." with something like "Muscle ultrasonography seems/appears be an effective tool for..."

Response (3)

We modified the sentence in Conclusion, as you recommended,

(Line 344) Muscle ultrasonography seems to be an effective tool for estimating muscle mass and screening for sarcopenia.

---

## [Editor Report · Decision Letter 2]

19 Dec 2022

PONE-D-22-16915R2Usefulness of muscle ultrasound in appendicular skeletal muscle mass estimation for sarcopenia assessmentPLOS ONE

Dear Dr. Kim,

Thank you for submitting your manuscript to PLOS ONE. After careful consideration, we feel that it has merit but does not fully meet PLOS ONE’s publication criteria as it currently stands. Therefore, we invite you to submit a revised version of the manuscript that addresses the points raised during the review process.

ACADEMIC EDITOR: Dear Authors, your manuscript has been revised by an expert in the field that found some minor issues.

We look forward to receiving your revised manuscript.

Kind regards,

Emiliano Cè

Academic Editor

PLOS ONE
---

## [Author Response · Author response to Decision Letter 2]

20 Dec 2022

We carefully checked all reference lists and found the #27 article did not correctly include the publication information. So, the information of reference #27 was updated in the revised manuscript as follows: 

(Line 455-458) 27. Monjo H, Fukumoto Y, Asai T, Ohshima K, Kubo H, Tajitsu H, et al. Changes in Muscle Thickness and Echo Intensity in Chronic Stroke Survivors: A 2-Year Longitudinal Study. J Clin Neurol. 2022;18(3):308-14. https://doi.org/10.3988/jcn.2022.18.3.308 PMID: 35196746

In addition, this manuscript (PONE-D-22-16915R3) added 1 more reference (Reference #10) to provide further explanations as the reviewer’s recommendation, compared to the previously revised manuscript (PONE-D-22-16915R1). Therefore, the final version of the revised manuscript (PONE-D-22-16915R3) included a total of 27 references. We have reviewed all reference list to ensure that it is complete and correct.

---

## [Editor Report · Decision Letter 3]

22 Dec 2022

Usefulness of muscle ultrasound in appendicular skeletal muscle mass estimation for sarcopenia assessment

PONE-D-22-16915R3

Dear Dr. Kim,

We’re pleased to inform you that your manuscript has been judged scientifically suitable for publication and will be formally accepted for publication once it meets all outstanding technical requirements.

Kind regards,

Emiliano Cè

Academic Editor

PLOS ONE
---

## [Editor Report · Acceptance letter]

5 Jan 2023

PONE-D-22-16915R3 

Usefulness of muscle ultrasound in appendicular skeletal muscle mass estimation for sarcopenia assessment 

Dear Dr. Kim:

I'm pleased to inform you that your manuscript has been deemed suitable for publication in PLOS ONE. Congratulations! Your manuscript is now with our production department. 

Kind regards, 

on behalf of

Professor Emiliano Cè 

Academic Editor

PLOS ONE